

# Drought propagation in the Rhine River basin and its impact on navigation using LAERTES-EU regional climate model dataset

Andrea L. Campoverde[1], Uwe Ehret[2], Patrick Ludwig[1], Joaquim G. Pinto[1]

[1]Institute of Meteorology and Climate Research, Troposphere Research (IMKTRO), Karlsruhe Institute of Technology (KIT), Karlsruhe, Germany
[2]Institute for Water and Environment (IWU), Karlsruhe Institute of Technology (KIT), Karlsruhe, Germany

*Correspondence to*: Andrea L. Campoverde (andrea.campoverde@kit.edu)

**Abstract.** Drought events have become more frequent in Europe over the past decades. The shipping and industrial sectors are severely affected by these events, e.g., due to significant reductions in water levels in the Rhine River and interruptions in the transport of goods. Hydrological droughts in the Rhine closely resemble extreme meteorological droughts identified using the Standardized Precipitation Evapotranspiration Index (SPEI) over both short and long periods. However, the possibility of determining low water level events using non-observed meteorological data, e.g., from large regional climate model datasets, and their implication on navigation has not yet been explored. The main objective of this study is to utilize the Large Ensemble of Regional Climate Model Simulations for Europe (LAERTES-EU) to search for extreme drought years, to assess their contribution to discharge, and to determine possible navigation restrictions on the waterways in the Rhine. We employed a methodology that evaluates the SPEI values to identify meteorological drought, which are then used to obtain discharge values by applying the hydrological model WRF-Hydro. The top 10 most extreme meteorological drought events in the LAERTES-EU dataset are considered in this study, assessing drought propagation by using the SPEI. These events displayed different degrees of severity in terms of duration and reduction of the streamflow as measured by the navigable threshold GlQ20 than the extreme drought observed in 2018. In the selected gauges, several LAERTES-EU events were ranked above 2018 when comparing them with the historical records in terms of mean discharge of the period June-November. These results imply that, even under today's climatic conditions, the streamflow values in the Rhine can be substantially worse than in 2018, generating costly economic and ecological consequences if mitigation measures are not implemented.

## 1. Introduction

A prolonged and strong negative imbalance in the water balance of a region is known as a hydrological drought event (Van Loon & Van Lanen, 2012). One crucial factor related to hydrological droughts to understand is drought propagation, which, according to Van Loon (2015), corresponds to the "change of drought signal as it moves through the terrestrial part of the hydrological cycle," meaning that not only precipitation should be considered but also soil moisture, groundwater, and



evapotranspiration. Van Loon (2015) explains that a combination of different aspects is required to develop a hydrological drought, such as a prolonged period of precipitation deficit, that hinders the water recharge and storage, an increase in potential evapotranspiration due to high temperatures or lack of moisture, aiding to a constriction of soil moisture depletion and production of local precipitation, and low groundwater levels, that directly contribute to discharge. The process of

drought propagation has been researched through the consideration of different drought indices (Standardized Precipitation Index (SPI), Standardized Precipitation and Evapotranspiration Index (SPEI), Standardized Streamflow Index (SSI)) and the various features of a hydro-meteorological drought, such as lag, intensity, attenuation, and lengthening (Edossa et al., 2010; Sattar et al., 2019; Van Loon, 2015; Sutanto et al., 2024; Li et al., 2024; Gu et al., 2020; Barker et al., 2016; Lin et al., 2023; Van Loon & Van Lanen, 2012; Liu et al., 2020; Wu et al., 2021). It has been found that not only is a meteorological analysis

crucial, but also the characteristics of the basins in terms of climate, catchment preconditions, and orography (Lin et al., 2023; Barker et al., 2016; Laaha et al., 2017; Bakke et al., 2020; Zhu et al., 2021). Van Loon and Lan Lanen (2012) presented a procedure to determine several types of hydrological droughts depending on the catchment characteristics and the climatological conditions. They conclude that extreme events highly differ whether the drought occurs in a basin with a snow season or in a basin where rivers have a pluvial regime. Erfurt et al. (2020) thoroughly categorized drought events in

the upper-middle section of the Rhine River in Germany using several indices. It was pointed out that it is necessary to include different indices because otherwise some events can be overlooked. Erfurt et al. (2020) reiterate the observation that there is no direct link between extreme meteorological and hydrological droughts, but it is the clustering of hydro-climatological components in the catchment that needs to be evaluated.

The impact of hydrological droughts differs depending on the type of basin where they occur. In some countries in Africa and Australia, where the rivers are only rainfed, extreme events, meaning prolonged hot and dry periods, directly affect people's daily lives (Sheffield & Wood, 2012). In parts of Europe, the consequences directly affect the economy by disrupting production and transport (Christodoulou et al., 2020), as well as the ecology in rivers, not only through the lower water levels but also through increased water temperatures that lead to algal blooms, fish mortality, and forced migration

from small rivers (ICPR, 2018). An analysis of the last 200 years of data concluded that around 30% of the most severe hydrological and meteorological droughts in the southwest of Germany have occurred since 2000 (Erfurt et al., 2020). The same study identified 2018 as the second most extreme event for the Rhine River, specifically from June to November. This position is the result of the analysis of hydrological records since 1900. The most extreme drought occurred in 1949, but given that the quality of the data has improved significantly in recent decades, we decided to focus on the drought of 2018,

also because it had a strong impact on navigation. The hydrological drought in 2018 event started with an early snow melt, followed by scarce rain in spring and summer (Rousi et al., 2023; Knutzen et al., 2025; Xoplaki et al., 2025), and the lowest recorded water levels in the middle section of the river (IKSR, 2020). Consequently, there was a significant interruption in shipping through the waterway, amounting to over 2 billion Euros in manufacturing losses (IKSR, 2020).





The Bundesanstalt für Wasserbau (BAW), the Federal Waterways Engineering and Research Institute in Germany, published a report on an action plan for low water events on the Rhine River (Bundesanstalt für Wasserbau, 2022). In their analysis, the BAW identified 16 locations along the river, between Iffezheim and Emmerich, where the minimum water level required for navigation is reduced by 30 percent. These 16 locations were classified as potential bottlenecks impacting river navigation during low water conditions. It was stated that to create a bottleneck, shallow water depths should not be the assumption, as there could still be enough fairway depth. However, the reduction of 30%, which was the case in 2018, is chosen because, based on the ship's characteristics, the discharge restricts the passage in both depth and width of the river (Bundesanstalt für Wasserbau, 2022). Therefore, given the limitations in time of observed information, using a regional climate dataset with more than a thousand years of information provides the opportunity to analyze possible events that could achieve conditions for transport interruptions. Vinke et al. (2022) stated that the significant reduction of streamflow in the Rhine River in 2018
led to an increase of three times more trips to maintain the transported volume under normal conditions, representing an additional cost for transportation of 5 million euros weekly by the end of the year. Therefore, assessing the impacts of extreme drought events on the waterways is imperative.

In this study, we evaluate a large ensemble of regional climate simulations under current climatic conditions(LAERTES-EU)
to identify extreme meteorological droughts that propagate to hydrological drought events and their subsequent impacts on navigation along the Rhine River. Therefore, the research focuses on the following key questions:

1.   Can the Standard Precipitation Evapotranspiration Index (SPEI) be used as a drought propagation indicator for the Rhine River catchment to facilitate the identification and extraction of hydrologically relevant events from the LAERTES-EU data set?
2.   Are there events in LAERTES-EU that are more severe than those observed?
3.   What is the impact on navigability of the Rhine River for selected extreme events from LAERTES-EU?

Addressing these questions will contribute to the study of waterways management under extreme drought conditions in the Rhine basin due to the costly maintenance of low-water corridors and changes in means of transportation.

**2. Materials and Methods**

**2.1 LAERTES-EU data set**

The Large Ensemble of Regional Climate Model Simulations for Europe data set (LAERTES-EU) (Ehmele et al., 2020) was generated in the scope of the MIKLIP project (Marotzke et al., 2016) by a series of simulations with the regional climate model Consortium for Small-scale Modeling Climate Model version 5 (COSMO-CLM5, hereafter CCLM5) that
dynamically downscale global climate model data from the Max Planck Institute for Meteorology - Earth System Model (MPI-ESM). The simulations varied according to the resolutions of the boundary conditions of MPI-ESM (Ehmele et al.,





2020). However, the setup and initialization of CCLM5 remain the same. The modeling process created more than 12.000 years of meteorological data, representing not chronological years but several members that were run for different periods (10 – 100 years) grouped in four blocks. The earliest simulated calendar year is 1900, and the latest hindcasts end in the calendar year 2028 (Ehmele et al., 2020). It was stated that when comparing LAERTES-EU precipitation with the observed data (E-OBS), the data blocks driven by high-resolution MPI-ESM showed higher correlations (Ehmele et al., 2022). The contrast was not based on absolute values but on spatial mean temporal variability. Ehmele et al. (2022) used the LAERTES-EU data to detect flood events and model the associated streamflow values along the Rhine River. The precipitation data had previously been bias corrected using the quantile mapping technique, which significantly reduced the bias throughout the basin except for the Alpine region, due to topographic variability (Ehmele et al., 2022). The authors also showed that the bias corrected data from blocks 2 and 4 were better correlated with the observed data. Block 4 was divided into two parts: the first part (4a) was forced with Coupled Model Intercomparison Project (CMIP5) specifications and has five members, and the second part (4b) with CMIP6 specifications and contains ten members. For this study, block 4b, containing 5900 model years, was selected due to the overall good correlation with observed data, the availability of the necessary variables for the hydrological model, and the extent of years (50% of the total data set).

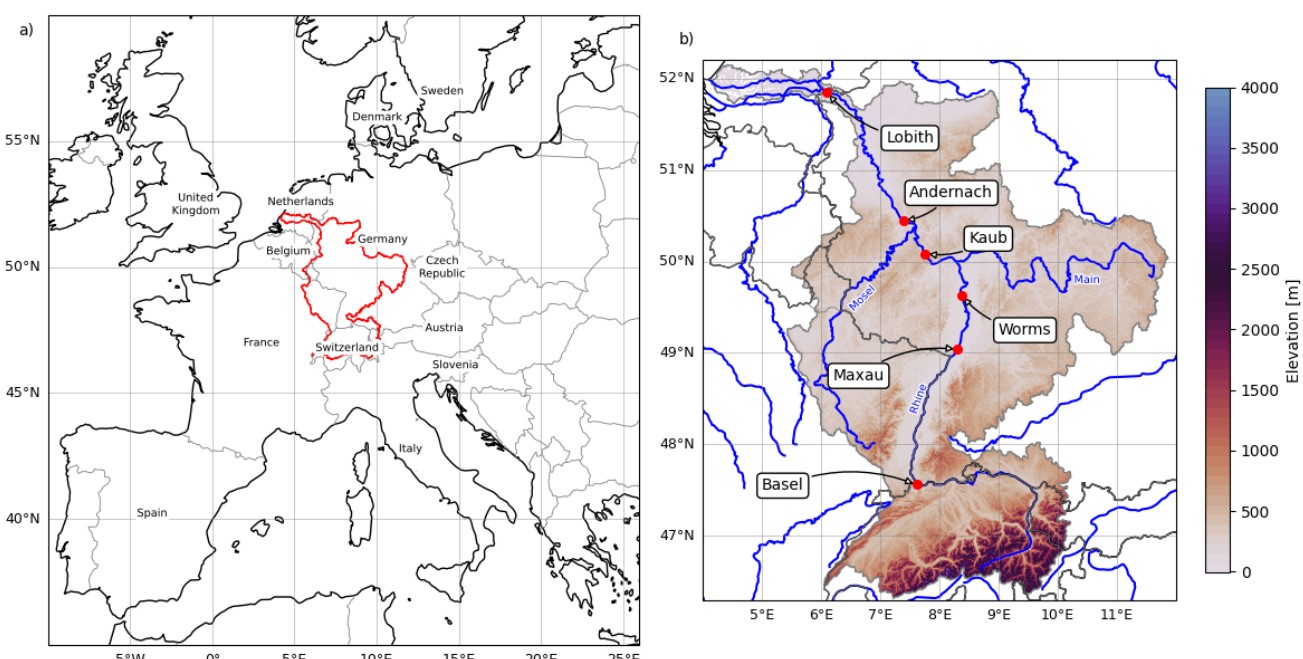

**Fig 1. Location of the Rhine River's catchment in Europe (a). Elevation in meters above sea level of the basin and hydrological gauge stations (red dots) that are part of the waterway of the Rhine used for shipment (b).**





## 2.2 Study Area

This study focuses on the Rhine River basin (Fig. 1). The extension of the catchment area is 9973 km$^2$, and the river's length is 1230 km, making it one of the longest rivers in Europe. The historic development of the Rhine Valley created a space for industry growth in several sectors. Therefore, one of the most important uses of the river is the transportation of different kinds of goods between Basel and Rotterdam (Table 1). (ICPR, https://www.iksr.org/en/topics/uses/industry/six-industrial-centres-along-the-rhine). BAW stated that when extreme hydrological drought occurs, several sections of the Rhine are affected by increased traffic of vessels and cargo boats in order to maintain the transported volume. They stated that six bottlenecks were detected around the gauge Kaub (Bundesanstalt für Wasserbau, 2022). The gauges selected for the hydrological analysis are Basel, Maxau, Worms, Kaub, Andernach, and Lobith (see Fig. 1b for the location of the gauges). With these gauges, the bottlenecks are considered, and the points where extremely low water levels have been recorded.

**Table 1. Industry sectors along the Rhine River valley. Source: International Commission for the Protection of the Rhine (ICPR).**

| Economic Centre | Type of Industry |
|---|---|
| Basel/Muhlhaus/Freiburg | Chemical, food, textile, metal |
| Strasbourg | Cellulose, food, textile, metal |
| Rhine-Neckar | Chemical |
| Frankfurt-Rhine-Main | Chemical, rubber, electrical, metal, services |
| Metropolitan region Rhine-Ruhr | Petrochemical, refinery, metal, car, services, trade |
| Rotterdam-Europoort | Shipyard, refinery, chemical, metal, car, services |

## 2.3 Standard Precipitation Evapotranspiration Index

Droughts can be categorized as meteorological, hydrological, agricultural, and socioeconomical, and their conditions have been studied using different methodologies, e.g., Knutzen et al. (2025) (Edossa et al., 2010; Sattar et al., 2019; Van Loon, 2015; Sutanto et al., 2024; Li et al., 2024; Gu et al., 2020; Barker et al., 2016; Lin et al., 2023; Van Loon & Van Lanen, 2012; Liu et al., 2020; Wu et al., 2021). They are often assessed based on observational data and the deviation from long-term mean values. However, one of the most challenging tasks is the analysis of the propagation from one type of drought to the other. Erfurt et al. (2020) analyzed the drought conditions in the southwestern state of Germany, Baden-Württemberg, of the last 200 years of hydro-meteorological and vegetation data to understand the dynamics through several indices. Furthermore, due to their interdisciplinary approach, their research gave insight into their similarities and correlations. According to Erfurt et al. (2020), hydrological drought events in the Rhine, from July to November (case of the hydrological drought of 2018), have similarities with the Standard Precipitation Evapotranspiration Index (SPEI) (Vicente-Serrano et al., 2010) of three, six, and twelve months, specifically of August, September, and December, respectively. Regarding their correlation analysis, the values were 0.54, 0.64, and 0.70, respectively. Their results show that it is more likely to find





extreme hydrological drought events by searching for extreme meteorological droughts using SPEI (Erfurt et al., 2020). Based on these results, SPEI3(Aug), SPEI6(Sep), and SPEI12(Dec) were selected as the first identifiers of meteorological drought in the LAERTES-EU data set.

The estimation of SPEI is based on the difference between the monthly sum of precipitation ($P_m$) and the monthly mean potential evapotranspiration ($PET_m$) (WB) (Eq. 4). This approach accounts for anomalies in precipitation and the impacts of temperature on evapotranspiration. For this study, the Thornthwaite method (Thornthwaite, 1948) was selected to calculate the PET as shown in Eq. 1, due to the variables available in the LAERTES-EU data set. In the equation $L$ is the mean day length, $N$ is the number of days in the month, $Ta$ is the daily air temperature, $I$ is heat index estimated from the annual sum of

the monthly mean temperature values (Eq. 2), and $a$ is a constant value calculated with I as shown in Eq. 3.

For SPEI, a logistic distribution was chosen to fit the WB monthly values of the period LAERTES-hist, consisting of 30 years of data obtained from mean precipitation and temperature values of block 4b. This fitting establishes the likelihood of occurrence for accumulation intervals of three, six, and twelve months during the historical period (see Eq. 5). The

distribution function requires the location ($\alpha$) and scale ($\beta$) parameters, which are estimated using the method of maximum likelihood. Here, the x value represents WB calculated with LAERTES-hist.

Subsequently, the cumulative distribution function of the logistic distribution function was applied using Eq. 6 to all of the 590 decadal ensemble members to determine the SPEI values. For this equation, x represents the WB values calculated with

the ensemble members. As a last step, a normal distribution quantile function was applied to the values obtained from the previous step. The entire process was performed using Python. The formulas to estimate the SPEI values and their components are provided below:

$$PET = 1.6 \left(\frac{L}{12}\right)\left(\frac{N}{30}\right)\left(\frac{10Ta}{I}\right)^a \tag{1}$$

$$I = \sum_{i=1}^{12}\left(\frac{Ta_i}{5}\right)^{1.514} \tag{2}$$

$$a = (6.75x10^{-7}) * I^3 - (7.71x10^{-5}) * I^2 + (1.792x10^{-2}) * I + 0.49239 \tag{3}$$

$$WB = P_m - PET_m \tag{4}$$

$$f(x;\alpha,\beta) = \left(e^{-\frac{x-\alpha}{\beta}}\right) * \beta \left(1 + e^{-\frac{x-\alpha}{\beta}}\right)^{-2} \tag{5}$$

$$F(x;\alpha,\beta) = 1 * \left(1 + e^{-\frac{x-\alpha}{\beta}}\right)^{-1} \tag{6}$$




### 2.4 WRF-Hydro Model

The hydrological model WRF-Hydro was selected to simulate the streamflow of the ten most extreme drought years identified in LAERTES-EU to determine if the meteorological drought leads to a hydrological drought. WRF-Hydro (Gochis et al., 2018) is a numerical hydrological model developed to improve the land-atmosphere interaction with the climate model

WRF (NCAR, 2024). It consists of a land surface model, Noah-MP, and routing schemes that simulate subsurface, overland, channel, baseflow, and lake/reservoir. These schemes establish the water contributions to the river channels and require that the hydrological parameters of the model be calibrated, which was achieved by Campoverde et al. (2025) for the Rhine River basin. The authors showed that with spatially distributed hydrological parameters and without the lake/reservoir scheme, the model is able to reproduce streamflow values during extreme drought conditions in the Rhine River basin (Campoverde et

al., 2025). The setup of Campoverde et al. (2025) is used in this study. The 10 most extreme years identified in LAERTES-EU based on the SPEI were used as input for the model. The required variables are at surface level: precipitation, air temperature, the components u and v from wind, specific humidity, long and shortwave radiation, and pressure. The complete year of each event was simulated, plus the preceding year as a spin-up period to provide realistic initial conditions for the drought year. Hydrological droughts are long-term and slow-evolving phenomena, so the model was operated in daily

(rather than hourly) resolution.

### 2.5 Navigation threshold GlQ20

The Central Commission for the Navigation of the Rhine (CCNR) has been establishing benchmarks for navigation limits at the gauges along the Rhine for the past 93 years. The equivalent discharge (GlQ20) is a long-term average based on 100-year

streamflow values and is recalculated every ten years (Zentral Kommission für die Rheinschifffahrt, 2014). This value is then used to calculate the equivalent water level (GlW20), which represents the water level that corresponds to 20 days of discharge below the long-term average for the Rhine River (Wasserstraßen- und Schifffahrtsverwaltung des Bundes, 2014). This water level serves as the minimum threshold at which navigation remains possible (Table 4) (Zentralkomission für die Rheinschifffahrt, 2022). BAW has also defined GlW20 as the benchmark to determine actions for maintenance performed by

the Waterways and Shipping Administration (WSV), thus facilitating the evaluation of fairway conditions (Federal Waterways Engineering and Research Institute, 2019). The threshold used in this project is from 2012, as it was used to evaluate the impacts of the drought event in 2018. To determine the GlQ20, CCNR utilized the period of 04/1911 to 03/2011 (Zentral Kommission für die Rheinschifffahrt, 2014).

Furthermore, the Federal Waterways Engineering and Research Institute (BAW) in Germany has established different threshold levels, according to the reduction of the GlQ20, where they have to ensure that the fairway has a minimum width





for nautical purposes (Bundesanstalt für Wasserbau, 2022). In this study, we used GlQ20 to analyze the severity of drought events in terms of affecting ship navigation on the Rhine.

## 2.6 Design of experiments

Due to limitations on computational time required by the hydrological model, ten extreme case studies were selected from LAERTES-EU. The selection process was based on a ranking system, ordering the cases from most extreme to least extreme. This procedure was performed on the individual SPEI3(Aug), SPEI6(Sep), and SPEI12(Dec). Afterwards, it was established that the cumulative ranking positions of the three indices provided the most extreme combination. It is worth noting that the extreme drought years in the first year of the decadal simulation were discarded due to the importance of 210 considering the basin conditions before the dry year.

The methodology was applied to observed data to test the combined SPEI ranking values for their ability to predict hydrological droughts. The SPEI values, corresponding to the domain of our project, were taken from the Global SPEI database (Beguería et al., 2023), and the mean streamflow values for the period of June to November were obtained from the 215 Global Runoff Data Center (GRDC, 2022). The mean discharge values are sorted in ascending order to establish a hydrological drought. Therefore, the first values correspond to the lowest streamflow values.

**Table 2. Ranking of the ten most extreme observed meteorological droughts using combined SPEI3(Aug), SPEI6(Sep), SPEI12(Dec), and hydrological droughts at the stations in the Rhine's waterway. Source: Global SPEI database (Beguería et al.,**
**2023), GRDC. Years with the same ranking in meteorological and hydrological are in bold, and the years where it differs are underlined.**

| Rank | Combined SPEI | Basel | Maxau | Worms | Kaub | Andernach | Lobith |
|------|------|------|------|------|------|------|------|
| 1 | **2018** | **2018** | **2018** | **2018** | **2018** | **2018** | 1976 |
| 2 | **2003** | **2003** | **2003** | **2003** | 1976 | 1976 | 2018 |
| 3 | **1976** | **1976** | **1976** | **1976** | 2003 | 2003 | 2003 |
| 4 | **2015** | **2015** | **2015** | **2015** | **2015** | **2015** | **2015** |
| 5 | 1991 | 1971 | 1971 | 1971 | 1971 | 1971 | 1989 |
| 6 | **1989** | **1989** | **1989** | 2011 | 2011 | 2011 | 1971 |
| 7 | 1971 | 2011 | 2011 | 1989 | 1989 | 1989 | 2011 |
| 8 | 1990 | 1972 | 2005 | 2005 | 1991 | 1991 | 1991 |
| 9 | **2009** | 2017 | **2009** | 1972 | 1972 | 2005 | 2005 |
| 10 | 1973 | 2005 | 2017 | 1991 | 2005 | 2009 | 2009 |





Each column in Table 2 provides the sorting of the observed SPEI and discharge values for each station. The numbers in
bold indicate a ranking match between the SPEI and at least one gauge. The underlined years mean that there is a mismatch
in the ranking between SPEI and hydrological years. If the year is not in bold or underlined, it means that there was no match
between the ranking of the SPEI and hydrological drought years in the first ten positions.

We can conclude, based on the results presented in Table 2, that by combining the ranking of SPEI3(Aug), SPEI6(Sep), and
SPEI12(Dec), we can find extremely low water events. In 6 events, the ranking of SPEI matches the position of the
hydrological drought events (bold). The SPEI years were ranked in different positions on the gauges in 7 different years
(underlined). Overall, this approach allows identifying meteorological droughts that can potentially become hydrological
drought events.

## 3. Results

### 3.1 Drought propagation

Using the SPEI for the accumulation periods of 3, 6, and 12 months to identify extreme meteorological drought events
within the LAERTES-EU revealed 229 cases that were simultaneously categorized as extreme events. From these, we
selected the ten most extreme events for further analysis.

Fig. 2 shows the results for the SPEI of all years of the LAERTES-EU block 4b data set, including the selected ten years
with color lines, which are named EV, followed by the number of their SPEI rank. It is noticeable that the chosen events
show more severity for SPEI-12, with events 2, 3, 4, and 5 having index values below -2 for the entire year, which reflects
the influence of the dry periods of the previous year. In contrast, for events 1 and 6-10, the SPEI-3 and SPEI-6 values show a
transition from wet conditions at the beginning of the year to dry conditions in summer and fall.






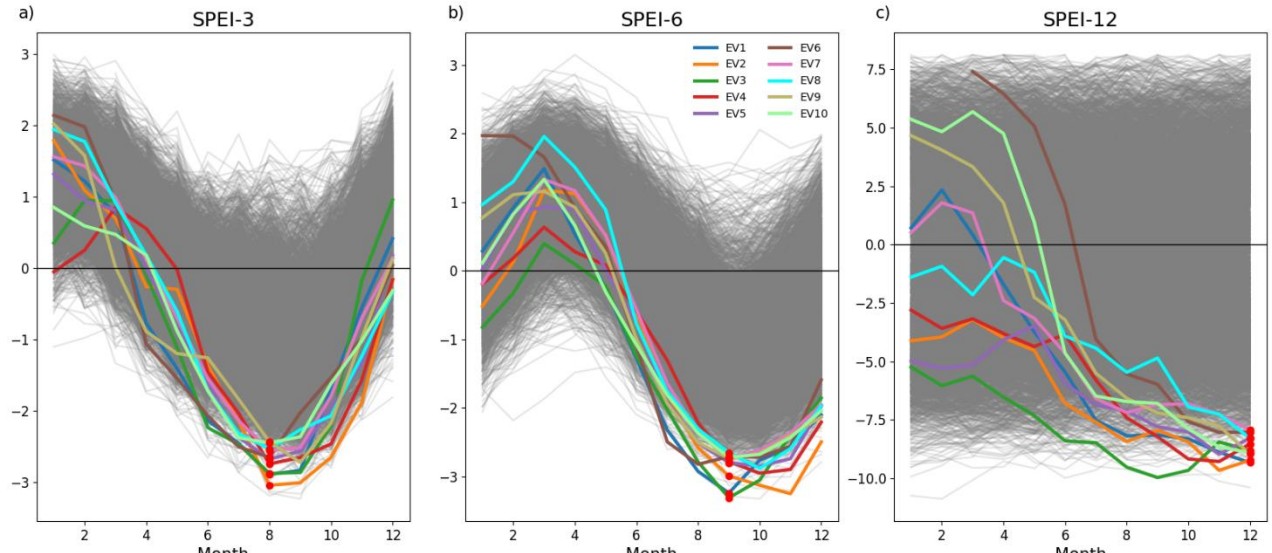

**Fig 2. SPEI values obtained from the evaluation of block 4b of LAERTES-EU for the accumulation periods of 3 (a), 6 (b), and 12 (c) months. The 10 most extreme years are displayed in color lines; all other years are shown in gray. The red dots are the minimum SPEI values used for the ranking of the extreme events.**

The SPEI values from observations (Fig. 3) of the drought years 2018 and 2003 display similar behavior compared to the selected LAERTES-EU events. Fig. 2a and 2b show a drop in the SPEI3 and SPEI6 values during the summer months and a recovery during autumn. It is essential to note that there cannot be a direct comparison between Fig.2 and Fig. 3 because there are two different datasets and reference periods to determine SPEI values. In LAERTES-EU, there is a decadal evaluation of 590 ensemble members (5900 years) of model-generated data from decadal hindcasts (i.e., initialized with

historical data). In contrast, the observed data from the Global SPEI database uses continuous historical meteorological information from the last 120 years. Furthermore, the large negative or positive values displayed in Figure 2c are due to the much larger ensemble and thus much larger variability in the regional climate simulations, which means that ensembles can potentially contain more extreme events. Hence, the ensemble member's water budget (P-PET) has a higher deviation from the estimated mean values in LAERTES-hist. The SPEI values show how many standard deviations an event differs from the

mean values. Therefore, the results show that the deviation of the selected events from the long-term mean of the surface water budget (P-PET) is significantly greater.





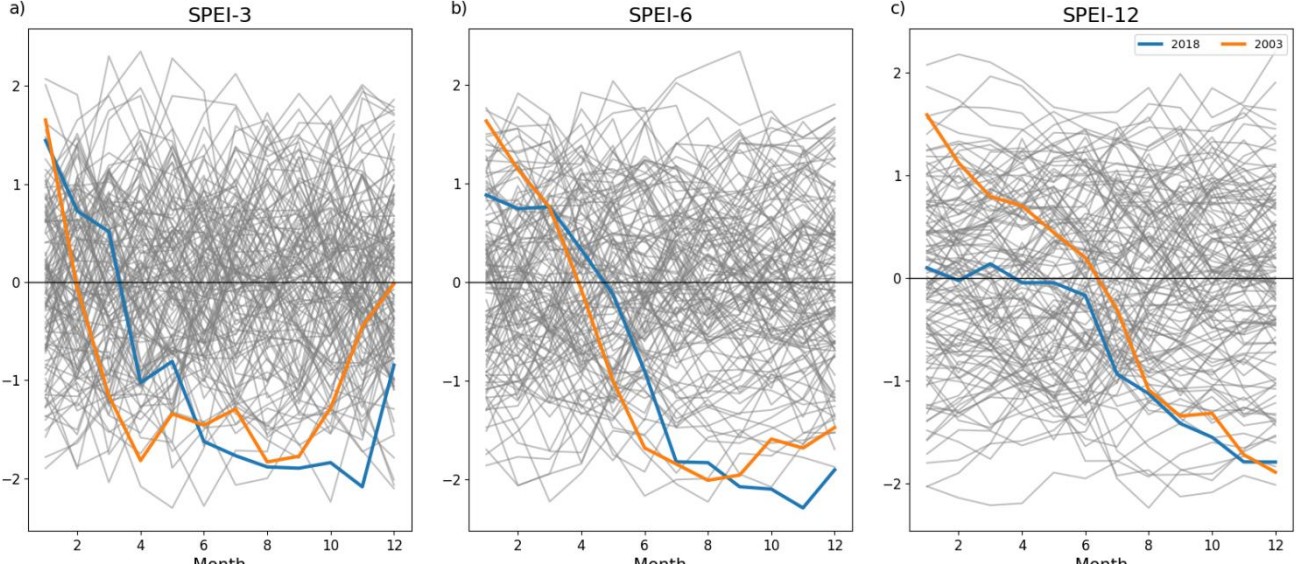

**Fig 3. SPEI values of the Rhein Basin domain from the Global SPEI database (1901-2022) (Beguería et al., 2023) In the accumulation period of 3 (a), 6 (b), and 12 (c) months. The two most recent extreme drought events (2003 and 2018) are displayed in color lines; all other years are shown in gray.**

Considering SPEI12, it is noticeable that at the beginning of the year, the SPEI12 values are above zero for both historical cases, which means that the previous year did not have dry conditions (Fig. 3c). A similar behaviour is shown in Fig. 2c where some events, namely, for EV1, EV6, EV7, and EV9. The other events, EV2, EV3, EV4, EV5, and EV8, display a different behaviour, having negative values from the beginning of the year, signaling water deficit preconditions, which are not mitigated but propagated throughout the year.

An anomaly analysis was also performed with the precipitation and temperature variables from block 4b in LAERTES-EU. The assessment corresponds to a difference between the average temperature of LAERTES-hist and the annual mean value of each LAERTES-EU year. The precipitation values from LAERTES-EU were added to obtain the annual mean, subtracted from the annual mean of LAERTES-hist, and later divided by the annual mean of LAERTES-hist. Fig. 4 shows that all the selected LAERTES-EU events occurred in the dry and hot years (red dots), and one at the limits of the distribution. The observed year 2018 was added to provide context for the behaviour of the LAERTES-EU events, showing that the annual anomalies for 2018 are in the same cluster. ERA5 was used to calculate the observed anomalies with a reference period of 1981-2010. The driest year within the LAERTES-EU years is not among the most extreme events, but one of the hottest years is. This demonstrates that temperature strongly influences the determination of SPEI extreme drought values for the LAERTES-EU dataset. The following section (Section 3.2) will show the results obtained from using the ten LAERTES-EU events as input for the hydrological model WRF-Hydro. The model output provides the streamflow values of the selected



years, and it will be demonstrated that these events are hydrological droughts. Therefore, we can conclude that a link has been established between meteorological droughts expressed by a combination of SPEI 3, 6, and 12 and hydrological droughts, which confirms the results from Erfurt et al. (2020).

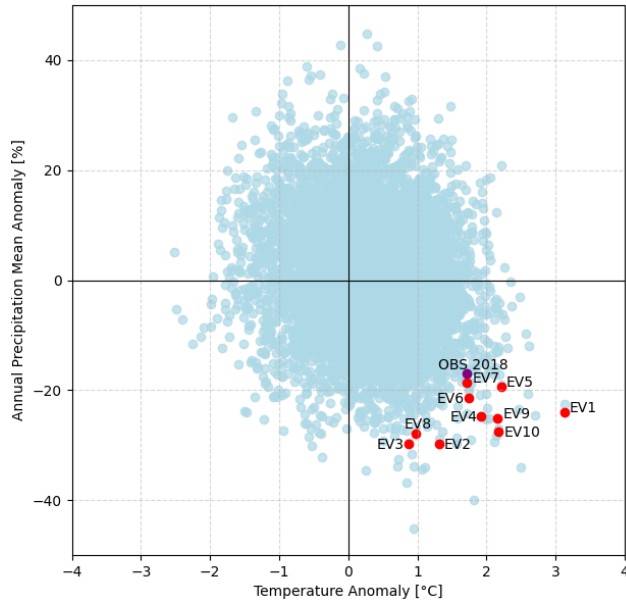

**Fig 4. Thermopliviogram showing annual anomalies of precipitation (%) and temperature (°C) for all simulated years from LAERTES-EU in block 4b, relative to the LAERTES-hist climatological mean. Blue dots represent all simulated years, red dots highlight the ten most extreme events, and the purple dot shows the observed year 2018, estimated using ERA5 data with a reference period of 1981-2010 (Hersbach et al., 2018).**

**3.2 Drought Severity**

The streamflow values simulated with WRF-Hydro based on the selected extreme events from LAERTES-EU and the historical data (1970-2018) obtained from the Global Runoff Data Center (GRDC) were analyzed to determine how severe the LAERTES-EU events are compared to the drought years on record. For both datasets, and separately for each year and each gauge, the mean streamflow values for the period June to November were calculated and combined into a single dataset. The dataset was then ordered by the magnitude, and the rank of each of the LAERTES-EU-based events within the

set was determined.





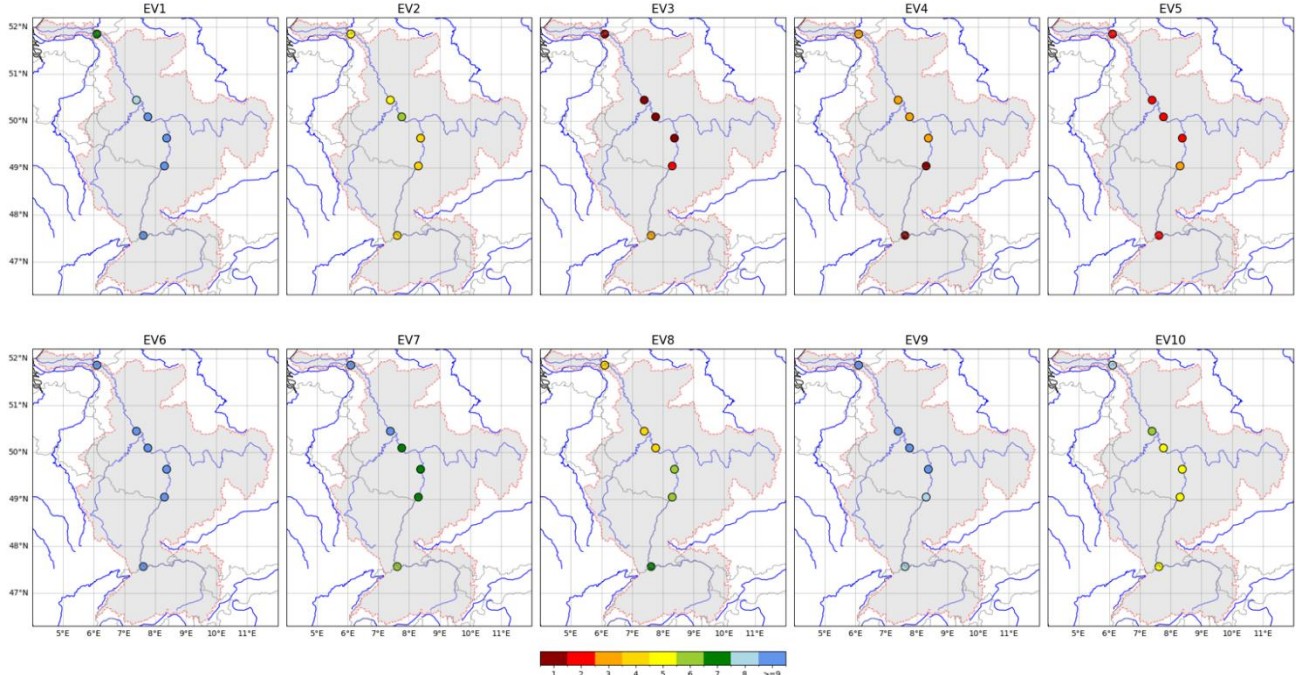

**Fig 5. Spatial distribution of the 10 most extreme modeled hydrological drought events identified using LAERTES-EU block 4 dataset, ranked against historical discharge observations from 1970 to 2018. Each panel (EV1-EV10) shows the event specific ranking across hydrological gauges (locations detailed in Fig. 1), with colors indicating drought severity ranks. Higher ranks (in red) correspond to the more extreme conditions.**

The results are displayed in Fig. 5, where a dot represents each selected gauge, and the color denotes the rank of the LAERTES-EU event (EV1-EV10). The gauges are categorized in the first seven positions in EV2, EV3, EV4, EV5, and EV8. This means the hydrological drought of 2018 has been displaced by several LAERTES-EU events (Fig. S10 and Table 3). However, the 2018 event is still featured in the top ten events, which confirms how extraordinary 2018 was.

Furthermore, Fig. 5 shows that for EV3, four out of six stations (Worms, Kaub, Andernach, and Lobith) rank first. For EV4 and EV5, the gauges are among the top three positions. Table 3 provides a detailed ranking of all the LAERTES-EU events and the observed drought years for each station (Fig. S11). It is clear that these three events represent the most severe droughts. Additionally, Table 3 indicates that the top six positions are held by the mean discharge values of the LAERTES-EU extreme events in all gauges, surpassing observed events. For the gauge Basel, all ten LAERTES-EU events are more severe compared to the observed ones. However, it is possible that more events in LAERTES-EU may continue to displace the 2018 drought event, because our analysis consists only of the top ten extreme meteorological events in block 4b of LAERTES-EU. The ranking analysis indicates that the drought year 2018 was extraordinary because it displaced several LAERTES-EU events in all gauges. Moreover, the simulation of 2018 using WRF-Hydro (Fig. S12) was also compared in the context of observed hydrological droughts (Fig. S11), and the ranking slightly varies only at the stations Andernach and



Lobith. Therefore, a comparison of the LAERTES-EU hydrological extreme drought years with the results from the simulation of the discharges is acceptable, and it is explained in Section 3.3.

**Table 3. Ranking of the mean streamflow values at the gauges between June and November of the LAERTES-EU in contrast with the observed hydrological data. Source of observed data: GRDC.**

| Rank | Basel | Maxau | Worms | Kaub | Andernach | Lobith |
|------|-------|-------|-------|------|-----------|--------|
| 1 | EV4 | EV4 | EV3 | EV3 | EV3 | EV3 |
| 2 | EV5 | EV3 | EV5 | EV5 | EV5 | EV5 |
| 3 | EV3 | EV5 | EV4 | EV4 | EV4 | EV4 |
| 4 | EV2 | EV2 | EV2 | EV8 | EV8 | EV8 |
| 5 | EV10 | EV10 | EV10 | EV10 | EV2 | EV2 |
| 6 | EV7 | EV8 | EV8 | EV2 | EV10 | $OBS_{1976}$ |
| 7 | EV8 | EV7 | EV7 | EV7 | **$OBS_{2018}$** | EV1 |
| 8 | EV9 | EV9 | **$OBS_{2018}$** | **$OBS_{2018}$** | EV1 | EV10 |
| 9 | EV1 | **$OBS_{2018}$** | EV1 | EV1 | EV6 | EV6 |
| 10 | EV6 | EV1 | EV6 | EV6 | EV7 | **$OBS_{2018}$** |
| 11 | **$OBS_{2018}$** | EV6 | EV9 | EV9 | $OBS_{1976}$ | EV7 |
| 12 | $OBS_{2003}$ | $OBS_{2003}$ | $OBS_{2003}$ | $OBS_{1976}$ | EV9 | EV9 |
| 13 | $OBS_{1976}$ | $OBS_{1976}$ | $OBS_{1976}$ | $OBS_{2003}$ | $OBS_{2003}$ | $OBS_{2003}$ |
| 14 | $OBS_{2015}$ | $OBS_{2015}$ | $OBS_{2015}$ | $OBS_{2015}$ | $OBS_{2015}$ | $OBS_{2015}$ |
| 15 | $OBS_{1971}$ | $OBS_{1971}$ | $OBS_{1971}$ | $OBS_{1971}$ | $OBS_{1971}$ | $OBS_{1989}$ |

Comparing the ranking of the hydrological (Table 3) with the meteorological events (Fig. 2, 5) reveals some differences. These differences are due to the preconditions of the basin. Fig. 2c provides the SPEI values for the accumulation period of

twelve months for every month of the year. When comparing the first five events (EV1-EV5), it is evident that EV1 shows positive values in January, whereas events two to five (EV2-EV5) exhibit negative values. This demonstrates that EV1 occurred under wet preconditions, while the other four events were subject to dry preconditions on the large time scales. Thus, the soil conditions of the previous year are also essential to understand the behavior of the river discharges and the propagation from meteorological to hydrological droughts. Fig. S13 displays the monthly mean soil moisture content across

the 2m soil column depth of the entire domain for the ten events and their preceding years. In this figure, events 2, 3, 4, and 5 show a significant decline in soil moisture during the summer months of the preceding year and recovered during winter, in contrast with the other events, where the soil moisture values do not show a drastic change in the same period.

## 3.3 Impacts on navigability

In this section, the hydrological model simulations are analyzed in terms of the impacts on navigation by means of the

equivalent discharge (GlQ20) threshold (see Section 2.6). A comparison was performed with the simulated discharge values





of the 2018 event, which ranks similar to the observed data as stated in Section 3.2 (Fig. S11-S12). This was done to have a direct contrast of the model output, where the configuration remains the same but the input varies. One was obtained from ERA5, and the other one from the LAERTES-EU dataset. The results are shown in Fig. 6, 7, and S1-S8. Fig. 6 displays the results from the EV3, which was categorized as the most severe event of the ten LAERTES-EU events, and shows lower

discharge values than those from 2018. For EV3, the duration of the drought period, in which the streamflow is below GlQ20, is, on average, 115 days among all stations (Fig. 8). This means that for navigation on the Rhine, EV3 is an even more severe event than the recorded year 2018, where on average the duration period below GlQ20 was 93 days (Fig. 8 dotted line).

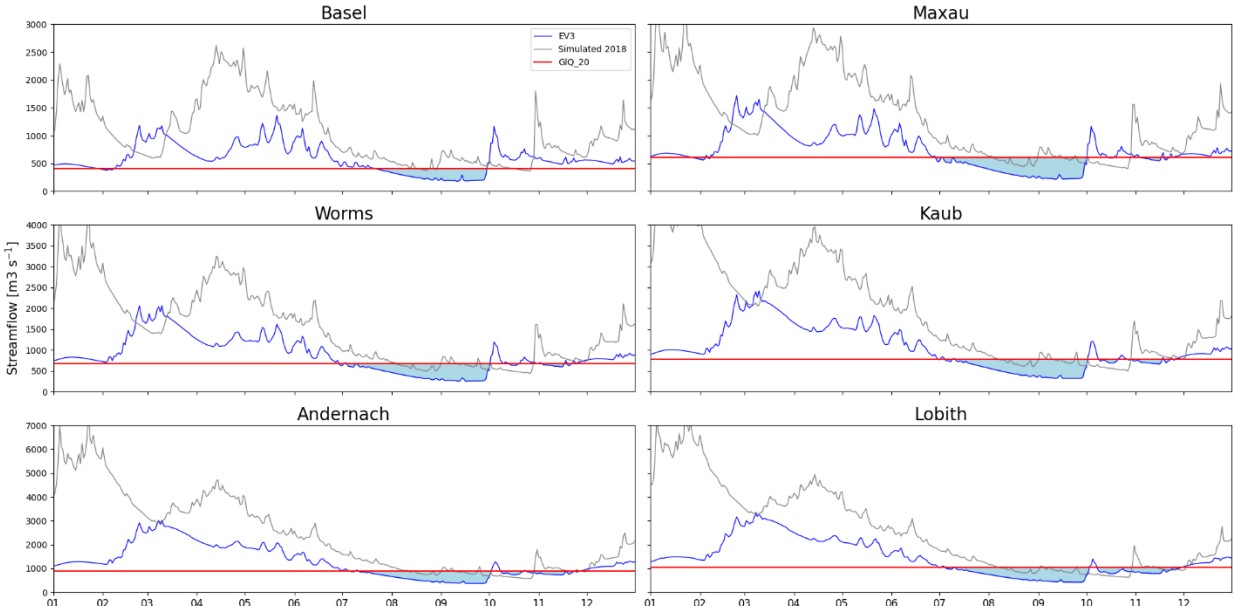

**Fig 6. Daily hydrographs at the gauges in the waterway of the Rhine River of the EV3 (blue line), in comparison with simulated hydrograph of the year 2018 (gray), and the navigation threshold GlQ20 (red).**

It is notable that the drought period extended into the following year for EV2. The hydrographs in Fig. 7 indicate that the discharge values below the GlQ20 are extended for approximately 45 days beyond December. The conditions from EV2 led

to a drought event that lasted over 50% of the hydrological year, making it an even more severe case for navigation impact, in terms of duration, than EV3.

Furthermore, the BAW proposed additional thresholds (GlQ20 -10%, GlQ20 -20%, and GlQ20 -30%) for assessing low water events (Bundesanstalt für Wasserbau, 2022). They concluded that the greater the reduction of GlQ20, the more severe

the event is, and the stricter the measures need to be applied because dredging will not be sufficient to maintain the fairway, such as a change in the type of transportation or a complete restriction on the circulation of boats. For example, according to



a BAW report, it was stated that the observed streamflow values in 2018 were below GlQ20 -30% (Bundesanstalt für Wasserbau, 2022). These boundaries were used to compare the output of WRF-Hydro to determine the severity with respect to navigability. Most events display a more than 30% reduction on at least one day. However, particularly at the stations Maxau and Worms, all events have days below 30% (Fig. 8). EV2 is shown as the most lasting and intense event among the ten analyzed in this project when taking into consideration the threshold GlQ20. In section 3.2, EV2 was not ranked as the most severe event because the analysis was an average streamflow value of a fixed period. Here, we are taking into consideration the entire duration of the drought. These results imply that in most of the events in this study, transportation interruptions are more significant than in 2018.

**Table 4. GlQ20 threshold levels for streamflow at different gauges on the Rhine River. Source: (Zentral Kommission für die Rheinschifffahrt, 2014).**

| Gauge | GlQ20 (m s$^{-1}$) |
|---|---|
| Basel | 488 |
| Maxau | 609 |
| Worms | 682 |
| Kaub | 784 |
| Andernach | 887 |
| Lobith | 1020 |

In addition to analyzing the discharge values, the duration of the selected extreme events is another aspect to consider. The number of days below the respective GlQ20 thresholds for each gauge was calculated (Fig. 8). Notably, EV2 includes the longest period of low water levels (GlQ20), exceeding 200 days for the stations Maxau, Worms, Kaub, and Lobith. It is important to note that at the Kaub station, most bottlenecks occur during low water events. Nine LAERTES-EU events show between 12 and 130 accumulated days when the discharge is lower than GlQ20 -30%. Additionally, by using the observed discharge data, we can see in Fig. S9 the number of days below GlQ20 -30% in 2018. The duration of the stations Worms, Kaub, Andernach, and Lobith was between 5 and 25 days. In contrast, Fig. 8 indicates that during the events from LAERTES-EU EV2, the same gauges experienced a duration approximately between 29 and 6 times longer than the observed event in 2018 (140-150 days). From Fig. 8, we can point out that there are cases where the discharges are severely low for more than 30 days, which represents a significant increase in shipping costs due to changes to small cargo boats and, in some cases, if the fairway solutions are too expensive, a complete interruption of transportation on the Rhine.





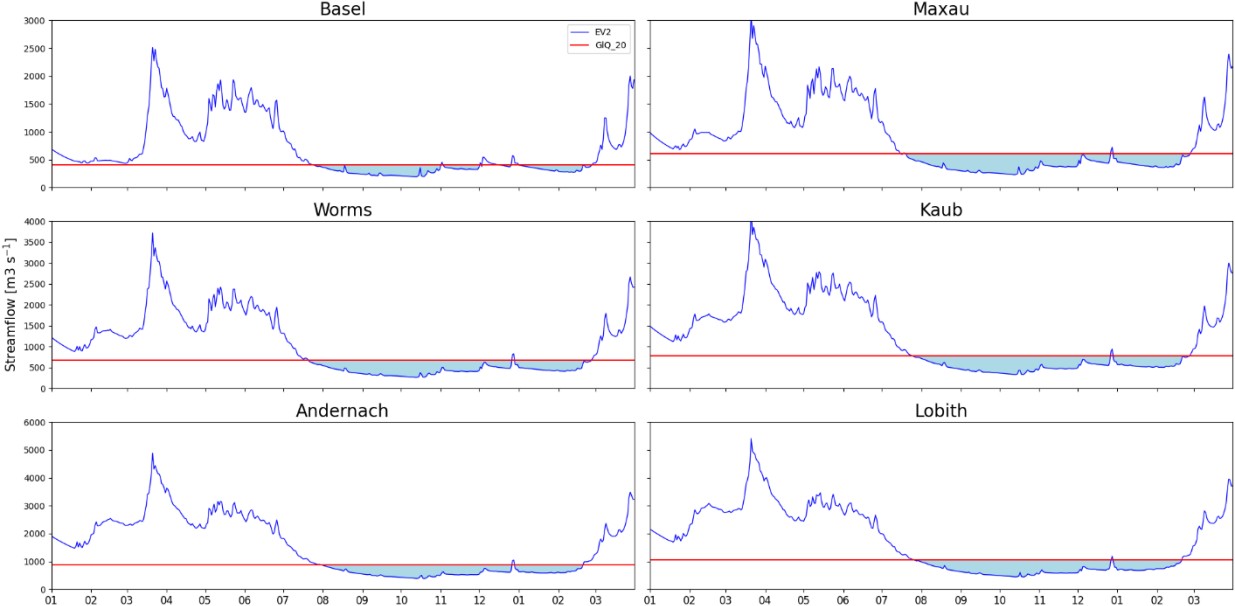

**Fig 7. Daily hydrographs at the gauges in the waterway of the Rhine River of the EV2 (blue line), in comparison with the navigation threshold GlQ20 (red).**

## 4. Summary and conclusions

Increasing drought periods and their impact on water levels in the Rhine River have caused costly consequences in the industry sector. Such events are difficult to predict due to their complexity. However, drought propagation has proven to be an appropriate tool for determining the relationship between meteorological and hydrological droughts (Gu et al., 2020; Van Loon & Van Lanen, 2012; Edosssa et al., 2010; Barker et al., 2016; Sattar et al., 2018; Sattar et al., 2019; Erfurt et al., 2020). This study focused on analyzing not yet observed meteorological drought scenarios in the Rhine basin under today's climatic conditions, also known as black swan events, from the LAERTES-EU dataset. The most severe detected events were used as input for the hydrological model WRF-Hydro to test how they translate into discharge values.

The correlation between meteorological and hydrological drought indices obtained by Erfuhrt et al. (2020) was used to scan the LAERTES-EU dataset, consisting of several thousand simulated years of regional climate model data under recent and current climate conditions, for extreme drought events. Given that it is not possible to simulate the entire LAERTES-EU dataset in the hydrological model WRF-Hydro, it was crucial to find a methodology that would both identify extreme drought events and determine whether these events translate into low water levels in the Rhine River. The selected events were simulated with WRF-Hydro, and the discharges were ranked among the observed droughts to assess their severity compared to historical extreme events. Lastly, an analysis of the impacts on navigation was performed to evaluate the different levels of restrictions the LAERTES-EU events would cause in shipping.





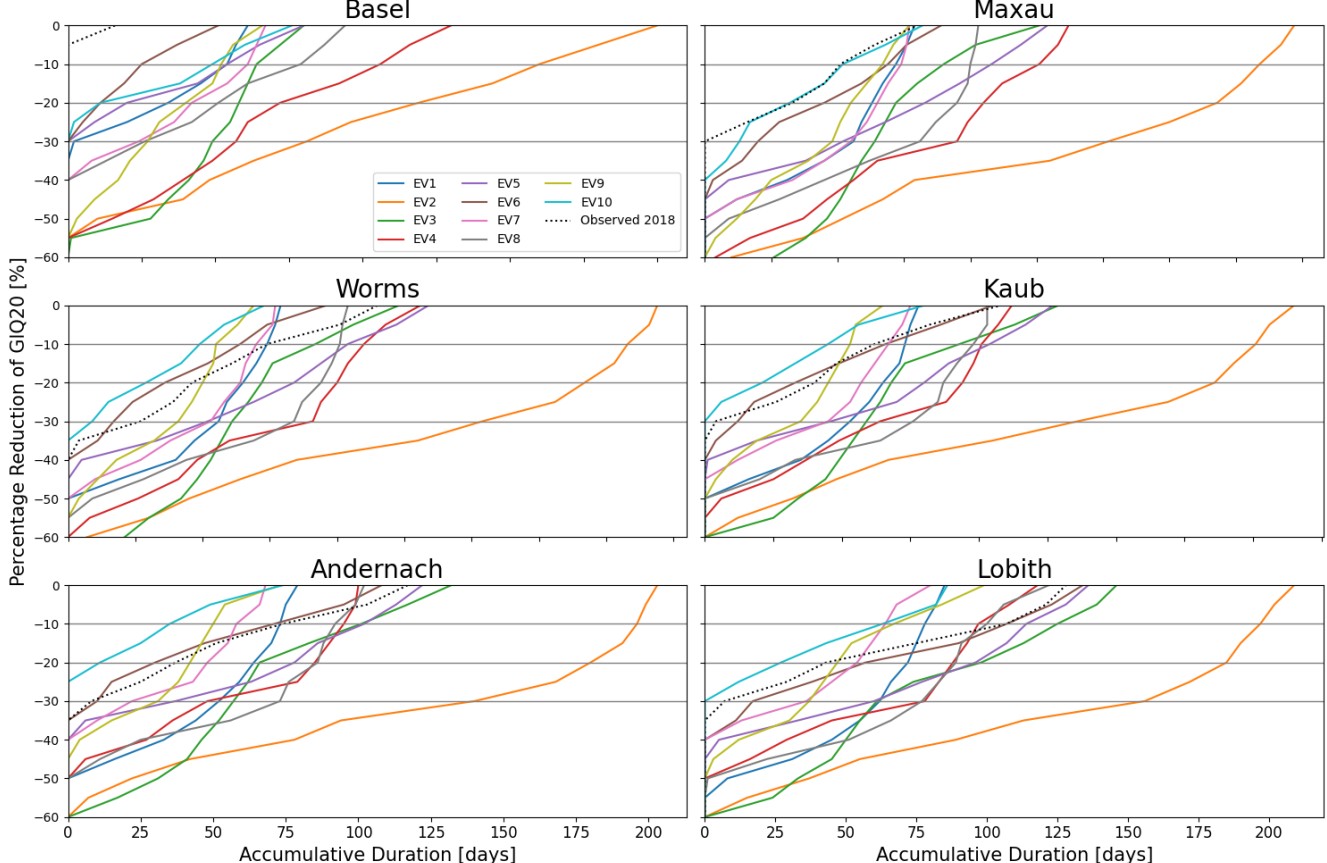

**Fig 8. Accumulated duration (in days) of streamflow reductions below the GlQ20 threshold at six gauges along the Rhine River (Basel, Maxau, Worms, Kaub, Andernach, and Lobith) for each of the 10 selected extreme events (EV1-EV10) and the observed 2018 hydrological drought (dotted line). The vertical axis shows the percentage reduction in discharge relative to GlQ20. Horizontal grid lines mark the reductions of 10%, 20%, and 30% below the GlQ20 threshold.**

The SPEI (Vicente-Serrano et al., 2010) has been widely applied for drought monitoring and assessment, where normally historical observed meteorological information is used (Sattar et al., 2018; Li et al., 2024; Gu et al., 2020; Rousi et al., 2023; Knutzen et al., 2025; Xoplaki et al., 2025). The results of the SPEI methodology proposed in our research indicate that for a dataset that contains decadal meteorological information, it is possible to estimate the SPEI by deconstructing the procedure, which consisted of obtaining a 30-year average precipitation and temperature value among all the decadal ensembles to train the logistic distribution. Fig. 2 shows a distinction between the wet and dry periods for the ensembles, and the results resemble the temporal variability of the SPEI values when compared to the observed values in Fig. 3. A ranking of meteorological droughts has been used in other studies; however, it is usually used for single SPEI values to indicate that the lowest values are the most extreme drought events (Gu et al., 2020; Li et al., 2024; Knutzen et al., 2025; Rousi et al., 2023). Taking into account this methodology and combining it with a collective ranking among three cumulative periods (SPEI3,





SPE6, SPEI12) with individual extreme drought values, we selected the ten most extreme meteorological drought events. Additionally, we demonstrated that these events translated into low-water-level events. The use of the three SPEI indices and

the translation into hydrological droughts aligns with the work of Erfurt et al. (2020), who reported the correlation between these indices in the Rhine using observed data.

By ranking the average simulated streamflow values for the period from June to November, we showed the categorization of the not-yet-observed LAERTES-EU selected years in comparison with the mean values of the same period for observed data

from GRDC. We found more severe events in LAERTES-EU than what has been observed in 2018. However, the meteorological drought ranking does not align with the ranking of the low water level events. This can be attributed to the fact that the more severe hydrological events (EV2, EV3, EV4, EV5, and EV8) occurred in the context of dry preconditions. These findings align with other studies that concluded that for various basins around the world, it is difficult to provide a direct link between meteorological and hydrological droughts due to the different components that lead to a significant

decrease in discharge (Edossa et al., 2010; Sattar et al., 2019; Van Loon, 2015; Sutanto et al., 2024; Li et al., 2024; Gu et al., 2020; Barker et al., 2016; Lin et al., 2023; Van Loon & Van Lanen, 2012; Sattar et al., 2018; Liu et al., 2020; Wu et al., 2021). Therefore, the results from a study in the Rhine basin that considered different drought indices (Erfurt et al., 2020) were imperative. It is important to note that Campoverde et al. (2025) stated that the outcome of the calibration of the hydrological model WRF-Hydro slightly overestimates the streamflow values during a drought event. Therefore, it is

possible that the LAERTES-EU events can be even more severe.

The navigation impact assessment revealed the varying effects of the LAERTES-EU events concerning duration and streamflow reduction in relation to the navigation threshold GlQ20. The results indicated that EV2 has the longest duration, lasting 45 days beyond the end of the year (Fig. 7). Additionally, the gauges Worms and Maxau show that for all the

considered LAERTES-EU events, the discharge values were reduced by at least 30% for a minimum of 10 days (Fig. 8). The GlQ20 -30% parameter is essential as this reduction of discharge caused bottlenecks, transport interruptions, and changes in the types of boats used at different sections of the Rhine during the drought in 2018. Therefore, our findings can potentially be utilized for mitigation analysis by providing results for the ten extreme cases that have not yet been observed, drawn from a 5900-year dataset. These results could guide costly mitigation solutions such as dredging or modifications to the type of

vessels or cargo boats. GlQ20 is a threshold that is updated every 10 years to account for current variations in streamflow values. In this study, we use the GlQ20 values established in 2012 because these values were utilized to analyze the 2018 event, the latest extreme drought event in the Rhine. Further evaluations should consider updated values.

The main conclusions from our study are as follows:

1.  SPEI values are suitable to predict the propagation of meteorological to hydrological droughts in the Rhine basin. In particular, selecting index values below -2 for the cumulative periods of three (SPEI3), six (SPEI6), and twelve





months (SPEI12), for August, September, and December, respectively. Extreme meteorological drought events selected by this method and used as forcing data for WRF-Hydro resulted in extreme low-water-level events throughout the selected years.

2. Using a ranking approach, in which the extreme events identified in LAERTES-EU years were compared to observed drought events, we concluded that some of the selected LAERTES-EU events were more severe than the observed 2018 drought. It is important to note that 2018 displaced a few of the LAERTES-EU events, demonstrating how extraordinary this drought year was. Additionally, EV3 has four out of six gauges ranked in first place.

3. The navigation thresholds of GlQ20 provided a perspective of how severely the shipping sector would be impacted in case one of the LAERTES-EU events would happen in reality. Some cases show discharge values that are significantly below the most extreme case (GlQ20 –30%), and in other cases, the duration of the event is longer than the most extreme case on record.

This study provides insights into not yet observed hydrological low-water-level cases that are more severe than events from historical observational data. Furthermore, we can conclude that the ranking of hydrological drought events does not strictly follow the ranking of meteorological events, because the preconditions of the basin have a significant impact on low water events. The selected events with a significant reduction in streamflow indicate the requirement for expensive mitigation solutions and increased shipping costs compared to those during the summer of 2018. Our findings provide realistic, not-yet-

observed cases for mitigation assessment during droughts at the Rhine waterway.

*Code and Data Availability.* The code of the WRF-Hydro (version 5.2.0) model is freely available online and can be downloaded from https://ral.ucar.edu/projects/wrf_hydro/model-code (last access: 16 July 2024). The large part of the LAERTES-EU data is available via the German Climate Computing Center (DKRZ). The observation data can be freely

downloaded from the Global Runoff Data Center of the Bundesantsalt für Gewässerkunde (BfG) https://portal.grdc.bafg.de/applications/public.html?publicuser=PublicUser#dataDownload/Home (GRDC, 2022) (last access: 19 December 2024)

*Author contributions.* The concept of the manuscript was developed by all authors. ALC pre-processed the input for WRF-

Hydro, performed simulations, analyzed data, and prepared the figures. UE contributed with methodology and data analysis. Funding was obtained by PL and JGP. ALC wrote the initial paper draft. All authors contributed to discussions, comments, and text revisions.

*Competing interests.* At least one of the (co-)authors is a member of the editorial board of Natural Hazards and Earth System Sciences.



*Acknowledgements*. The spatial distribution of WRF-Hydro's hydrological parameters was done with the aid of Thomas Rummler, who thankfully provided a Python script that extracts the different parameters and transforms them into tiff files. This work used resources of the Deutsches Klimarechenzentrum (DKRZ) granted by its Scientific Steering Committee (WLA) under project 983.

*Financial support*. The article processing and charges for this open-access publication were covered by the Karlsruhe
Institute of Technology (KIT). ALC was partially funded by a grant from the Climate Research and the Center for Disaster Management and Risk Reduction Technology (CEDIM). JGP and ALC thank the AXA research fund for the support.

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
