# Peer review of "Drought propagation in the Rhine River basin and its impact on navigation using LAERTES-EU regional climate model dataset"

_EGUsphere, 2025_

## Author Comment (AC1)

**Request:**

This study assessed hydrological droughts in the Rhine basin using model output simulations and a hydrological model. The purpose was to evaluate the ability of atmospheric drought indices to capture hydrological drought variability across the large Rhine basin. Although the topic of the proposal is of great scientific and applied management interest, I identify significant data and methodological issues. The focus of the project is not clearly defined, and there are substantial uncertainties in the experimental design—particularly regarding the hydrological modelling, which is poorly detailed, the use of model outputs, and their comparability with observations. Moreover, the manuscript lacks a meaningful discussion section. Further details on methodological uncertainties are provided below.

**Response:**

Thank you for your comment. In the manuscript, the model calibration description was not detailed because a manuscript under review was cited. Currently, the publication is published and available at this link <a href="https://doi.org/10.1080/15715124.2025.2581608">https://doi.org/10.1080/15715124.2025.2581608</a>, We hope this document resolves any questions regarding the model's description and calibration. We will include the citation in a revised version of the manuscript. We would also like to clarify that our main objective with the manuscript is to utilize the Large Ensemble of Regional Climate Model Simulations for Europe (LAERTES-EU) to search for extreme drought years, to assess their contribution to discharge, and to determine possible navigation restrictions on the waterways in the Rhine, as it was stated in the lines (L13-L16).

Also, in a revised version of the manuscript, we will add a discussion section

**Request:**

Line 145. The method used to calculate atmospheric evaporative demand is overly simplistic and uncertain. If the available data do not allow the application of the Penman–Monteith equation, the appropriate alternative would be the Hargreaves equation, which uses maximum and minimum temperature.

**Response:**

We appreciate your concern regarding the method used to estimate Potential Evapotranspiration. We acknowledge that the Penman-Monteith equation is considered the standard when comprehensive meteorological data is available. However, the LAERTES-EU dataset contains daily meteorological variables. Therefore, we will maintain that the Thornthwaite method is appropriate for this study, given our data constraints. We will add a more detailed description of the reason to use Thornthwaite in the manuscript.

**Request:**

Lines 161–170. These formulations are unnecessary and confusing, as they merge the calculation of potential evaporation with the fitting of data to compute the SPEI.

**Response:**

Thank you for your comment. In a revised version of the manuscript, we will separate the equations to provide a better understanding and clearer separation of the calculations.

**Request:**

Section 2. This section is confusing, and it is not possible to infer a logical structure in the methodological design. It remains unclear how the various elements described—simulations from GCMs and RCMs, the drought index, and hydrological modelling—are integrated into a coherent methodological framework aligned with the stated objectives. Overall, the hydrological modelling component is not sufficiently informative. Planning a hydrological model for such a large basin is an enormous task, subject to major uncertainties, and these challenges and methodological limitations are not adequately addressed in the manuscript's methodology section.

**Response:**

We appreciate your comment. To improve understanding of the process from data to analysis, we will add the figure below. In it, a description of the partitions of the LAERTES-EU data set is provided, with emphasis on the blocks that were bias-corrected. The process to determine the SPEI values is in green. The method for selecting the top 10 events is in the purple boxes. Finally, the actions to establish the effects of the selected drought events on the Rhine River's discharge are shown in orange blocks. This information will be added to a revised version of the manuscript.

Fig 2. Overview of the data and modeling workflow in the study. Description of the LAERTES-EU data blocks and their content (on the left in light blue). Flowchart indicating the process to obtain the Potential Evapotranspiration (PET) and the Standard Precipitation and Evapotranspiration Index (SPEI) (in green). The selection process of the top 10 most extreme drought events in the LAERTES-EU dataset is represented in purple boxes, and the analysis of their impacts on navigation is shown in orange.

Regarding the model description and calibration process, the manuscript cited a scientific publication that was under review. Additionally, we submitted the final draft for the reviewers as additional information. Since our submission, this manuscript has been published and can be found at <a href="https://doi.org/10.1080/15715124.2025.2581608">https://doi.org/10.1080/15715124.2025.2581608</a>. As this was an extensive work, we kindly ask that you refer to this publication to resolve any questions.

**Request:**

A key issue is whether the assessment based on climate models accounts for the fact that these models do not reproduce the natural climate variability observed in reality. This must be considered when comparing observations with model simulations, as in Figures 2 and 3. The authors explain that LAERTES-EU is a model-generated dataset from decadal hindcasts (i.e., initialized with historical data), but this does not guarantee that model simulations reproduce the observed variability. This is an essential point that must be clarified, since if the model simulations do not match observed variability, they cannot be reliably used in this study, which aims to evaluate drought propagation based on real drought events. For example, forcing the hydrological model with data that do not reflect the observed variability may introduce substantial bias into the conclusions, as the most significant drought events are identified from observations.

**Response:**

We appreciate your comment. Regarding the climate variability of the LAERTES-EU dataset, we noted in Section 2.1 that the work on the dataset, specifically the bias correction (BC), has significantly improved the spatial mean temporal variability, not the absolute variables (Figure below) (Ehmele et al., 2022). It is important to distinguish that the objective of LAERTES-EU was not to recreate observed data but to create never-before-seen events within the current climate.

Spatially mean monthly precipitation for blocks 2 and 4 in LAERTES-EU for uncorrected (uncorr) and bias-corrected (BC) data in comparison with two observed data E-OBS and HYRAS. Source: Ehmele et al. (2022)

In Figures 2 and 3, we showed that even with the not-observed events, they behave similarly to the recorded extreme drought events. Moreover, the evaluation of the drought propagation was done using a methodology from Erfurt et al. (2020) that was proven to work in the same area of study, where they found the correlation between SPEI 3, 6, and 12 with the hydrological droughts. Therefore, we are not comparing directly to observed events but on how to determine that these events occurred in the Rhine River basin. Our main objective is to demonstrate that if these events were to occur, what would be the impacts on discharge and, subsequently, on navigation. We are aware that we cannot compare the streamflow values to observed values

because these events have not occurred, but we can assess how extreme these drought events are under current norms and thresholds.

There is an explanation of these results in L250-L261: "The SPEI values from observations (Fig. 3) of the drought years 2018 and 2003 display similar behavior compared to the selected LAERTES-EU events. Fig. 2a and 2b show a drop in the SPEI3 and SPEI6 values during the summer months and a recovery during autumn. It is essential to note that there cannot be a direct comparison between Fig. 2 and Fig. 3 because there are two different datasets and reference periods to determine SPEI values. In LAERTES-EU, there is a decadal evaluation of 590 ensemble members (5900 years) of model-generated data from decadal hindcasts (i.e., initialized with historical data). In contrast, the observed data from the Global SPEI database uses continuous historical meteorological information from the last 120 years. Furthermore, the large negative or positive values displayed in Figure 2c are due to the much larger ensemble and thus much larger variability in the regional climate simulations, which means that ensembles can potentially contain more extreme events. Hence, the ensemble member's water budget (P-PET) has a higher deviation from the estimated mean values in LAERTES-hist. The SPEI values show how many standard deviations an event differs from the mean values. Therefore, the results show that the deviation of the selected events from the long-term mean of the surface water budget (P-PET) is significantly greater."

**Request:**

Indeed, Table 3 clearly shows that the simulations using model-generated data overestimate the severity of drought events. However, it is not possible to determine whether this problem arises from the input data or from the inherent uncertainties associated with modelling such a large basin.

**Response:**

Thank you for your comment. The contrast in Table 3 has two purposes: one is to show that the events found in LAERTES-EU are more severe than the ones on record, and the second is to highlight how extraordinary the 2018 drought event was. Therefore, it is not an overestimation of the severity of the events; it is that under current standards, the LAERTES-EU events are more extreme than the ones on record. This hypothesis was expected to occur due to the fact that the timeseries of LAERTES is much longer than observed records, and therefore the probability is high that it includes events more extreme than the observed ones. We will explain this in the discussion section of a revised version of the manuscript.

**References:**

Ehmele, F., Kautz, L.-A., Feldmann, H., He, Y., Kadlec, M., Kelemen, F. D., Lentink, H. S., Ludwig, P., Manful, D., & Pinto, J. G.: Adaptation and application of the large LAERTES-EU regional climate model ensemble for modeling hydrological extremes: A pilot study for the Rhine basin. Nat. Hazards Earth Syst. Sci., 22(2), 677–692. https://doi.org/10.5194/nhess-22-677-2022, 2022

Erfurt, M., Skiadaresis, G., Tijdeman, E., Blauhut, V., Bauhus, J., Glaser, R., Schwarz, J., Tegel, W., & Stahl, K.: A multidisciplinary drought catalogue for southwestern Germany dating back to 1801. Nat. Hazards Earth Syst. Sci., 20(11), 2979–2995. https://doi.org/10.5194/nhess-20-2979-2020, 2022